

# Probing Chern number by opacity and topological phase transition by a nonlocal Chern marker

Paolo Molignini[1], Bastien Lapierre[2], Ramasubramanian Chitra[3] and Wei Chen[4⋆]

1 T.C.M. group, Cavendish Laboratory, University of Cambridge,
19 J J Thomson Avenue, Cambridge, CB3 0HE, United Kingdom
2 Department of Physics, University of Zurich,
Winterthurerstrasse 190, 8057 Zurich, Switzerland
3 Institute for Theoretical Physics, ETH Zurich, 8093 Zurich, Switzerland
4 Department of Physics, PUC-Rio, Rio de Janeiro 22451-900, Brazil

⋆ wchen@puc-rio.br

## Abstract

In 2D semiconductors and insulators, the Chern number of the valence band Bloch state is an important quantity that has been linked to various material properties, such as the topological order. We elaborate that the opacity of 2D materials to circularly polarized light over a wide range of frequencies, measured in units of the fine structure constant, can be used to extract a spectral function that frequency-integrates to the Chern number, offering a simple optical experiment to measure it. This method is subsequently generalized to finite temperature and locally on every lattice site by a linear response theory, which helps to extract the Chern marker that maps the Chern number to lattice sites. The long range response in our theory corresponds to a Chern correlator that acts like the internal fluctuation of the Chern marker, and is found to be enhanced in the topologically nontrivial phase. Finally, from the Fourier transform of the valence band Berry curvature, a nonlocal Chern marker is further introduced, whose decay length diverges at topological phase transitions and therefore serves as a faithful indicator of the transitions, and moreover can be interpreted as a Wannier state correlation function. The concepts discussed in this work explore multi-faceted aspects of topology and should help address the impact of system inhomogeneities.

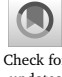

# 1  Introduction

Two-dimensional (2D) time-reversal (TR) breaking systems has been an important subject in the research of topological materials, and also one of the earliest systems discovered to have nontrivial topological properties [1]. The topological invariant in these systems is described by the Chern number, which is usually detected experimentally via measuring the quantized Hall conductance [2] that in practice is attributed to the formation of edge states. The bulk band structure, on the other hand, shows the same gapped energy spectrum in both the topologically trivial and nontrivial phases, and therefore bulk measurements not involving edge states are often not considered feasible to identify the Chern number. However, the theoretical proposal of the Chern marker and other real space constructions changed this paradigmatic viewpoint [3–13]. Through expressing the Chern number in terms of the projectors into the filled and empty bands, the Chern marker is introduced as a local quantity that is well-defined everywhere in the bulk, and recovers the Chern number in the clean, thermodynamic limit. In fact, it has been shown that such a real space formalism of topological invariant is also possible in other dimensions and symmetry classes [14–20]. It was further recognized recently that the Chern number and Chern marker are related to the circular dichroism of the 2D material [21–25], and so is the Berry curvature that integrates to the Chern number [26, 27], pointing to the possibility of directly detecting the topological order by means of bulk optical measurements.

In this paper, we introduce a number of new theoretical concepts related to Chern numbers and markers and show that relatively simple optical experimental protocols at finite temperature can be devised to directly access bulk state topology and topological criticality. We first introduce the Chern number spectral function defined as the optical conductivity of the 2D material under circularly polarized light divided by frequency [28]. The Chern number is then given by the frequency integration of this spectral function. We then show that this can be readily detected from the opacity (measured in units of the fine structure constant) of the 2D material to circularly polarized light, as previously done in graphene [29, 30]. This experimental protocol is then generalized to individual lattice sites, leading to the conclusion that the spectral function of the Chern marker can be extracted from the absorption power under circularly polarized light on each site. Additionally, we show that its rich internal spatial structure is represented by a Chern correlator, which is nothing but the nonlocal current-current correlator. The magnitude of the correlator is found to increase as the system enters topologically nontrivial phases, suggesting that the inherent fluctuations associated with the Chern number become more dramatic in the nontrivial phases. Finally, to characterize the quantum criticality near topological phase transitions (TPTs), we introduce the nonlocal Chern marker. This is defined from the off-diagonal elements of the Chern operator, in contrast to the usual Chern

marker defined from the diagonal element of the same operator. We see that this nonlocal Chern marker is equivalent to a previously proposed correlation function that measures the overlap between Wannier functions localized at different home cells [31–33], and becomes long-ranged as the system approaches TPTs, owing to the diverging Berry curvature in momentum space.

## 2 Linear response theory of local Chern marker

### 2.1 Local Chern marker in terms of Wannier states

As our work primarily explores the link between Chern markers and putative experimental measurables, we first review the original derivation of the Chern marker by Bianco and Resta [7], with an emphasis on the relation to Wannier states in the homogeneous case. We use the following notations for the bands and Bloch states: indices $n$ and $m$ denote the valence and conduction bands respectively, while $\ell$ denotes both bands. We denote $|\ell^{\mathbf{k}}\rangle$ as the periodic part of the Bloch state, and $|\psi_{\ell\mathbf{k}}\rangle$ as the full state that is related to the former by $\langle\mathbf{r}|\psi_{\ell\mathbf{k}}\rangle = e^{i\mathbf{k}\cdot\mathbf{r}}\langle\mathbf{r}|\ell^{\mathbf{k}}\rangle$. In the homogeneous and thermodynamic limit, the Bloch state of each band $|\ell^{\mathbf{k}}\rangle$ defines a Wannier state $|\mathbf{R}\ell\rangle$ by

$$|\ell^{\mathbf{k}}\rangle = \sum_{\mathbf{R}} e^{-i\mathbf{k}\cdot(\hat{\mathbf{r}}-\mathbf{R})}|\mathbf{R}\ell\rangle, \qquad |\mathbf{R}\ell\rangle = \sum_{\mathbf{k}} e^{i\mathbf{k}\cdot(\hat{\mathbf{r}}-\mathbf{R})}|\ell^{\mathbf{k}}\rangle, \qquad (1)$$

where $\hat{\mathbf{r}}$ is the position operator, $\mathbf{R}$ is a Bravais lattice vector, and the normalization factors are ignored for simplicity. The Wannier function at position $\mathbf{r} = (x,y)$ is given by $\langle\mathbf{r}|\mathbf{R}\ell\rangle = W_\ell(\mathbf{r}-\mathbf{R})$, which localizes around the home cell $\mathbf{R}$ in real space.

For a general model that contains $N_-$ valence bands, the periodic part of the fully antisymmetric valence band state at momentum $\mathbf{k}$ is given by

$$|n^{\mathrm{val}}(\mathbf{k})\rangle = \frac{1}{\sqrt{N_-!}}\epsilon^{n_1 n_2...n_{N_-}}|n_1^{\mathbf{k}}\rangle|n_2^{\mathbf{k}}\rangle...|n_{N_-}^{\mathbf{k}}\rangle, \qquad (2)$$

where $\epsilon^{n_1 n_2...n_{N_-}}$ is the fully antisymmetric Levi-Civita symbol. The Berry curvature of this valence band state in Eq. (2) on the $xy$-plane of a 2D system is given by [34]

$$\Omega_{xy}(\mathbf{k}) \equiv i\langle\partial_x n^{\mathrm{val}}|\partial_y n^{\mathrm{val}}\rangle - i\langle\partial_y n^{\mathrm{val}}|\partial_x n^{\mathrm{val}}\rangle = \sum_n \left[i\langle\partial_x n^{\mathbf{k}}|\partial_y n^{\mathbf{k}}\rangle - i\langle\partial_y n^{\mathbf{k}}|\partial_x n^{\mathbf{k}}\rangle\right], \qquad (3)$$

where $\partial_\mu \equiv \partial/\partial k_\mu$. Note that the Berry curvature is additive in the curvatures of each individual valence band. We rewrite the corresponding Chern number $\mathcal{C}$ defined from the momentum integration of $\Omega_{xy}(\mathbf{k})$ as [7]

$$\mathcal{C} = \int\frac{d^2\mathbf{k}}{(2\pi)^2}\Omega_{xy}(\mathbf{k}) = \sum_{n,m}\int\frac{d^2\mathbf{k}}{(2\pi)^2}i\langle\partial_x n^{\mathbf{k}}|m^{\mathbf{k}}\rangle\langle m^{\mathbf{k}}|\partial_y n^{\mathbf{k}}\rangle - (x\leftrightarrow y). \qquad (4)$$

In the following, we extensively use the identity linking the non-Abelian Berry connection to the charge polarization of the eigenstates and Wannier states [7,31,32,35–37]

$$i\langle m^{\mathbf{k}}|\partial_x n^{\mathbf{k}}\rangle = \langle\psi_{m\mathbf{k}}|\hat{x}|\psi_{n\mathbf{k}}\rangle/\hbar = \frac{1}{\hbar}\sum_{\mathbf{R}}e^{i\mathbf{k}\cdot\mathbf{R}}\langle\mathbf{0}m|\hat{x}|\mathbf{R}n\rangle = \frac{1}{\hbar}\sum_{\mathbf{R}}e^{-i\mathbf{k}\cdot\mathbf{R}}\langle\mathbf{R}m|\hat{x}|\mathbf{0}n\rangle, \quad \forall\, m\neq n, \qquad (5)$$

where $\hat{x}$ is the $x$-component of the position operator $\hat{\mathbf{r}}$ in real space. Using Eq. (5), $\mathcal{C}$ can be expressed in terms of the Wannier states [7]:

$$
\begin{aligned}
\mathcal{C} &= \sum_{nm} \int \frac{d^2\mathbf{k}}{(2\pi)^2} \frac{i}{\hbar^2} \langle \psi_{n\mathbf{k}} | \hat{x} | \psi_{m\mathbf{k}} \rangle \langle \psi_{m\mathbf{k}} | \hat{y} | \psi_{n\mathbf{k}} \rangle - (x \leftrightarrow y) \\
&= \frac{1}{a^2} \sum_{nm} \int \frac{d^2\mathbf{k}}{(2\pi\hbar/a)^2} \int \frac{d^2\mathbf{k}'}{(2\pi\hbar/a)^2} i \langle \psi_{n\mathbf{k}} | \hat{x} | \psi_{m\mathbf{k}'} \rangle \langle \psi_{m\mathbf{k}'} | \hat{y} | \psi_{n\mathbf{k}} \rangle - (x \leftrightarrow y) \\
&= \frac{1}{a^2} \sum_{nm} \sum_{\mathbf{R}} i \langle \mathbf{0}n | \hat{x} | \mathbf{R}m \rangle \langle \mathbf{R}m | \hat{y} | \mathbf{0}n \rangle - (x \leftrightarrow y) \\
&= \frac{1}{a^2} \sum_{nm} \sum_{\mathbf{R}} i \int d\mathbf{r} \int d\mathbf{r}' x_{\mathbf{r}} W_n^*(\mathbf{r}) W_m(\mathbf{r}-\mathbf{R}) y_{\mathbf{r}'} W_m^*(\mathbf{r}'-\mathbf{R}) W_n(\mathbf{r}') - (x \leftrightarrow y),
\end{aligned}
\tag{6}
$$

where the second line is valid because the matrix elements for $\mathbf{k} \neq \mathbf{k}'$ vanish. In terms of the projection operators to the valence and conduction band states

$$
\hat{P} = \sum_n \int \frac{d^2\mathbf{k}}{(2\pi\hbar/a)^2} |\psi_{n\mathbf{k}}\rangle\langle\psi_{n\mathbf{k}}|, \qquad \hat{Q} = \sum_m \int \frac{d^2\mathbf{k}'}{(2\pi\hbar/a)^2} |\psi_{m\mathbf{k}'}\rangle\langle\psi_{m\mathbf{k}'}|,
\tag{7}
$$

the Chern number can be recast as

$$
\mathcal{C} = \frac{i}{Na^2} \text{Tr}\left[ \hat{P}\hat{x}\hat{Q}\hat{y} - \hat{P}\hat{y}\hat{Q}\hat{x} \right] = \frac{i}{Na^2} \text{Tr}\left[ \hat{P}\hat{x}\hat{Q}\hat{y}\hat{P} - \hat{P}\hat{y}\hat{Q}\hat{x}\hat{P} \right],
\tag{8}
$$

where Tr denotes the trace over all the degrees of freedom on the lattice, and it is known that in practice an additional projector $\hat{P}$ has to be added to get the correct Chern marker locally on each site [7], as introduced below. In fact, this extra projector can also be derived from a universal topological marker constructed from spectrally flattened Dirac Hamiltonians [20]. The operator $\hat{\mathcal{C}} \equiv i[\hat{P}\hat{x}\hat{Q}\hat{y}\hat{P} - \hat{P}\hat{y}\hat{Q}\hat{x}\hat{P}]$ will be referred to as the Chern operator.

The Chern marker is introduced by considering a 2D tight-binding Hamiltonian $H = \sum_{\mathbf{r}\mathbf{r}'\sigma\sigma'} t_{\mathbf{r}\mathbf{r}'\sigma\sigma'} c_{\mathbf{r}\sigma}^\dagger c_{\mathbf{r}'\sigma'}$ with eigenstates satisfying $H|E_l\rangle = E_l|E_l\rangle$. In terms of $|E_l\rangle$, the projectors in Eq. (7) take the form

$$
\hat{P} = \sum_n |E_n\rangle\langle E_n|, \qquad \hat{Q} = \sum_m |E_m\rangle\langle E_m|.
\tag{9}
$$

The local Chern marker is defined by [7]

$$
\mathcal{C}(\mathbf{r}) = \frac{1}{a^2} \sum_\sigma \langle \mathbf{r}, \sigma | \hat{\mathcal{C}} | \mathbf{r}, \sigma \rangle \equiv \frac{i}{a^2} \langle \mathbf{r} | \left[ \hat{P}\hat{x}\hat{Q}\hat{y}\hat{P} - \hat{P}\hat{y}\hat{Q}\hat{x}\hat{P} \right] | \mathbf{r} \rangle.
\tag{10}
$$

In the following sections, we will present a formalism that generalizes both the Chern number $\mathcal{C}$ and the Chern marker $\mathcal{C}(\mathbf{r})$ to finite temperature, introduce their spectral functions, and suggest a simple optical measurement to detect them in contrast to traditional Hall conductance measurements.

## 2.2 Opacity measurement of finite temperature Chern number

The connection between the zero temperature Chern number and circular dichroism has been discussed in multiple works [21–25]. Here, we discuss a finite temperature generalization of these ideas, using a formalism similar to the derivation of Faraday effects in solids [38]. Via the finite temperature Chern number spectral function that represents a distribution of Chern number in energy, we show that the Chern number can be simply measured from the opacity of the 2D material to circularly polarized light. We define the current operators $\hat{j}_{c1} = \hat{j}_x + i\hat{j}_y$

and $\hat{j}_{c2} = \hat{j}_x - i\hat{j}_y$ at momentum $\mathbf{k}$ where $\hat{j}_\mu(\mathbf{k}) = e\partial_\mu H(\mathbf{k})$ with $e$ the electron charge and $H(\mathbf{k})$ the unperturbed single-particle Hamiltonian. These currents are relevant to the two circularly polarized oscillating electric fields $\mathbf{E}^{c1}(t) = (\hat{\mathbf{x}} + i\hat{\mathbf{y}})E_0 e^{-i\omega t}$ and $\mathbf{E}^{c2}(t) = (\hat{\mathbf{x}} - i\hat{\mathbf{y}})E_0 e^{-i\omega t}$. Due to minimal coupling $\delta H = -\mathbf{j}\cdot\mathbf{A}$ and $\mathbf{E} = -\partial\mathbf{A}/\partial t$, the two polarizations induce the corresponding perturbations: $\delta H^{c1}(\mathbf{k}) = \hat{j}_{c1} iE_0 e^{-i\omega t}/\omega$ and $\delta H^{c2}(\mathbf{k}) = \hat{j}_{c2} iE_0 e^{-i\omega t}/\omega$. Within linear response theory, the polarization induced currents satisfy $\langle\hat{j}_{c2}\rangle = \sigma_{c2,c1}(\mathbf{k},\omega)E_0 e^{-i\omega t}$ and $\langle\hat{j}_{c1}\rangle = \sigma_{c1,c2}(\mathbf{k},\omega)E_0 e^{-i\omega t}$, where the optical conductivity $\sigma$ is

$$\sigma_{c1,c2}(\mathbf{k},\omega) = \sum_{\ell<\ell'} \frac{\pi}{a^2\hbar\omega} \langle\ell|\hat{j}_{c1}|\ell'\rangle\langle\ell'|\hat{j}_{c2}|\ell\rangle \left[f(\varepsilon_\ell^{\mathbf{k}}) - f(\varepsilon_{\ell'}^{\mathbf{k}})\right] \delta\left(\omega + \frac{\varepsilon_\ell^{\mathbf{k}}}{\hbar} - \frac{\varepsilon_{\ell'}^{\mathbf{k}}}{\hbar}\right). \tag{11}$$

Note, $\sigma_{c2,c1}(\mathbf{k},\omega)$ is given by the same expression with $\hat{j}_{c1} \leftrightarrow \hat{j}_{c2}$. Here $|\ell\rangle \equiv |\ell^{\mathbf{k}}\rangle$ is the periodic part of the Block state, and the index $\ell$ enumerates both the valence and conduction band states, since at finite temperature both of them contribute to the Chern number, and $f(\varepsilon_\ell^{\mathbf{k}})$ is the Fermi distribution of the eigenenergy $\varepsilon_\ell^{\mathbf{k}}$. Note that the $\delta$-function in Eq. (11) with $\omega > 0$ ensures $\varepsilon_\ell^{\mathbf{k}} < \varepsilon_{\ell'}^{\mathbf{k}}$, such that it only accounts for the optical absorption process, as denoted by the notation $\sum_{\ell<\ell'}$. To proceed, we introduce the Berry curvature spectral function at momentum $\mathbf{k}$ by

$$\Omega_{xy}^d(\mathbf{k},\omega) = \sum_{\ell<\ell'} \left[i\langle\partial_x\ell|\ell'\rangle\langle\ell'|\partial_y\ell\rangle - (x\leftrightarrow y)\right]\left[f(\varepsilon_\ell^{\mathbf{k}}) - f(\varepsilon_{\ell'}^{\mathbf{k}})\right]\delta\left(\omega + \frac{\varepsilon_\ell^{\mathbf{k}}}{\hbar} - \frac{\varepsilon_{\ell'}^{\mathbf{k}}}{\hbar}\right), \tag{12}$$

which has been derived from a linear response theory [27]. When integrated over frequency, this function recovers the Berry curvature in Eq. (3) in the zero temperature limit where $\ell \to n$ and $\ell \to m$. Using Eq. (11) and $\langle\ell|\hat{j}_\mu|\ell'\rangle = e\left(\varepsilon_\ell^{\mathbf{k}} - \varepsilon_{\ell'}^{\mathbf{k}}\right)\langle\partial_\mu\ell|\ell'\rangle$, we find the following relation

$$\sigma_{c2,c1}(\mathbf{k},\omega) - \sigma_{c1,c2}(\mathbf{k},\omega) = 2\frac{\pi e^2}{a^2}\hbar\omega\,\Omega_{xy}^d(\mathbf{k},\omega). \tag{13}$$

Further integration over momentum [28]

$$\begin{aligned}
\sigma_{c2,c1}(\omega) - \sigma_{c1,c2}(\omega) &= \int \frac{d^2\mathbf{k}}{(2\pi\hbar/a)^2}\left[\sigma_{c2,c1}(\mathbf{k},\omega) - \sigma_{c1,c2}(\mathbf{k},\omega)\right] \\
&= \frac{2\pi e^2}{\hbar}\omega\int \frac{d^2\mathbf{k}}{(2\pi)^2}\Omega_{xy}^d(\mathbf{k},\omega) \equiv \frac{2\pi e^2}{\hbar}\omega\,\mathcal{C}^d(\omega),
\end{aligned} \tag{14}$$

defines what we call the Chern number spectral function $\mathcal{C}^d(\omega)$. Physically, $\mathcal{C}^d(\omega)$ can be interpreted as a "density of states" of the Chern number $\mathcal{C}^d$ indicating which eigenstates contribute the most to the Chern number, as we shall see in the following sections using concrete examples. In our notation, the superscript $d$ stands for "dressed" to indicate that it is a finite temperature generalization of the Chern number.

To relate the spectral function in Eq. (14) to experimental observables, we write the real part of the circularly polarized oscillating fields and the currents they induce as

$$\begin{aligned}
\mathbf{E}^{c1}(\omega,t) &= E_0(\hat{\mathbf{x}} + i\hat{\mathbf{y}})\cos\omega t, \\
\mathbf{j}_{c2}(\omega,t) &= \sigma_{c2,c1}(\omega)\mathbf{E}^{c1*}(\omega,t) = \sigma_{c2,c1}(\omega)E_0(\hat{\mathbf{x}} - i\hat{\mathbf{y}})\cos\omega t, \\
\mathbf{E}^{c2}(\omega,t) &= E_0(\hat{\mathbf{x}} - i\hat{\mathbf{y}})\cos\omega t, \\
\mathbf{j}_{c1}(\omega,t) &= \sigma_{c1,c2}(\omega)\mathbf{E}^{c2*}(\omega,t) = \sigma_{c1,c2}(\omega)E_0(\hat{\mathbf{x}} + i\hat{\mathbf{y}})\cos\omega t,
\end{aligned} \tag{15}$$

where $E_0$ is the strength of the field. The absorption power at each circular polarization is then given by

$$\begin{aligned}
W_a^{c1}(\omega) &= \langle\mathbf{j}^{c2}(\omega,t)\cdot\mathbf{E}^{c1}(\omega,t)\rangle_t = \sigma_{c2,c1}(\omega)E_0^2, \\
W_a^{c2}(\omega) &= \langle\mathbf{j}^{c1}(\omega,t)\cdot\mathbf{E}^{c2}(\omega,t)\rangle_t = \sigma_{c1,c2}(\omega)E_0^2,
\end{aligned} \tag{16}$$

where the time average gives $\langle \cos^2 \omega t \rangle_t = 1/2$. On the other hand, the incident power of the light per unit cell area of each polarization is $W_i = c\,\varepsilon_0 E_0^2 |\hat{\mathbf{x}} \pm i\hat{\mathbf{y}}|^2/2 = c\,\varepsilon_0 E_0^2$, so the difference in opacity for the two polarizations is $\mathcal{O}^{c1}(\omega) - \mathcal{O}^{c2}(\omega) = \left[ W_a^{c1}(\omega) - W_a^{c2}(\omega) \right]/W_i$, which can be used to extract the Chern number spectral function and subsequently the Chern number by

$$\mathcal{C}^d(\omega) = \frac{1}{8\pi\omega}\left[\frac{\mathcal{O}^{c1}(\omega) - \mathcal{O}^{c2}(\omega)}{\pi\alpha}\right], \qquad \mathcal{C}^d = \int_0^\infty d\omega\, \mathcal{C}^d(\omega), \tag{17}$$

where $\alpha = e^2/4\pi c\,\varepsilon_0 \hbar$ is the fine structure constant [29]. In other words, the Chern number spectral function can be simply extracted from the opacity difference between the two circular polarizations, similar to measurements in graphene [29, 30]. Moreover, because the finite temperature Chern number $\mathcal{C}^d$ is the frequency integrated spectral function, Eq. (17) implies that the opacity difference under circularly polarized light divided by frequency and then integrated over frequency must be a quantized integer at zero temperature, thereby realizing a topology induced frequency sum rule for noninteracting 2D materials [28]. This simple experimental protocol is easily accessible, thereby permitting a direct verification of the concepts proposed in our work.

We remark that the proper definition of Chern number at finite temperature has been contentious. Previous works based on linear response theory of DC Hall conductance suggest to define the finite temperature Chern number as the momentum-integration of the product of the Fermi distribution and the filled band Berry curvature $\sigma_{xy}^{DC} = \int \frac{d^2\mathbf{k}}{(2\pi)^2} \sum_n \Omega_{xy}^n f(\varepsilon_n^{\mathbf{k}})$, which is what is measured in transport experiments [39, 40]. In contrast, our formalism based on optical Hall conductivity in Eq. (11) leads to an expression that contains the difference between the Fermi distributions of the filled bands and those of the empty bands, and a matrix element involving both filled and empty bands stemming from the optical absorption process. Thus our finite temperature formalism differs from that of the DC Hall conductance, and is specifically formulated to describe the opacity measurement of the Chern number at finite temperature.

## 2.3 Linear response theory of finite temperature Chern marker

The finite temperature Chern number can further be written into real space using the formalism in Sec. 2.1, yielding

$$\begin{aligned}
\mathcal{C}^d &= \frac{a^2}{\hbar^2} \int \frac{d^2\mathbf{k}}{(2\pi\hbar/a)^2} \sum_{\ell<\ell'} \left[ i\langle \partial_x \ell | \ell' \rangle \langle \ell' | \partial_y \ell \rangle - (x \leftrightarrow y) \right]\left[ f(\varepsilon_\ell^{\mathbf{k}}) - f(\varepsilon_{\ell'}^{\mathbf{k}}) \right] \\
&= \frac{1}{a^2} \int \frac{d^2\mathbf{k}}{(2\pi\hbar/a)^2} \sum_{\ell<\ell'} \left[ i\langle \psi_\ell^{\mathbf{k}} | \hat{x} | \psi_{\ell'}^{\mathbf{k}} \rangle \langle \psi_{\ell'}^{\mathbf{k}} | \hat{y} | \psi_\ell^{\mathbf{k}} \rangle - (x \leftrightarrow y) \right]\left[ f(\varepsilon_\ell^{\mathbf{k}}) - f(\varepsilon_{\ell'}^{\mathbf{k}}) \right] \\
&= \frac{1}{a^2} \int \frac{d^2\mathbf{k}}{(2\pi\hbar/a)^2} \int \frac{d^2\mathbf{k}'}{(2\pi\hbar/a)^2} \sum_{\ell<\ell'} \left[ i\langle \psi_\ell^{\mathbf{k}} | \hat{x} | \psi_{\ell'}^{\mathbf{k}'} \rangle \langle \psi_{\ell'}^{\mathbf{k}'} | \hat{y} | \psi_\ell^{\mathbf{k}} \rangle - (x \leftrightarrow y) \right]\left[ f(\varepsilon_\ell^{\mathbf{k}}) - f(\varepsilon_{\ell'}^{\mathbf{k}'}) \right] \\
&= \frac{1}{Na^2} \sum_{\ell<\ell'} \left[ i\langle E_\ell | \hat{x} | E_{\ell'} \rangle \langle E_{\ell'} | \hat{y} | E_\ell \rangle - (x \leftrightarrow y) \right]\left[ f(E_\ell) - f(E_{\ell'}) \right] \\
&= \frac{1}{Na^2} \sum_{\ell<\ell'} \mathrm{Tr}\left[ i\hat{x} S_{\ell'} \hat{y} S_\ell - (x \leftrightarrow y) \right]\left[ f(E_\ell) - f(E_{\ell'}) \right], \tag{18}
\end{aligned}$$

where $|E_\ell\rangle$ is a lattice eigenstate obtained from diagonalizing the lattice Hamiltonian $H|E_\ell\rangle = E_\ell|E_\ell\rangle$, and we denote its projector by $S_\ell = |E_\ell\rangle\langle E_\ell|$. Here $|\psi_\ell^{\mathbf{k}}\rangle$ is the full Bloch state satisfying $\langle \mathbf{r}|\psi_\ell^{\mathbf{k}}\rangle = e^{i\mathbf{k}\cdot\mathbf{r}}\langle \mathbf{r}|\ell\rangle$, and in deriving Eq. (18) we have used

$$\int \frac{d^2\mathbf{k}}{(2\pi\hbar/a)^2} |\psi_\ell^{\mathbf{k}}\rangle\langle\psi_\ell^{\mathbf{k}}| = \sum_\ell S_\ell\,,$$
$$\int \frac{d^2\mathbf{k}}{(2\pi\hbar/a)^2} |\psi_\ell^{\mathbf{k}}\rangle\langle\psi_\ell^{\mathbf{k}}| f(\varepsilon_\ell^{\mathbf{k}}) = \sum_\ell S_\ell f(E_\ell)\,. \tag{19}$$

At zero temperature, the Fermi distribution becomes a step function hence the indices $\ell \to n$ and $\ell' \to m$ are limited to the valence and conduction bands, respectively, so the Chern number in Eq. (18) recovers the zero temperature results in Eqs. (6) and (8).

However, from the discussion after Eq. (8), an extra projector analogous to $P$ must be added to Eq. (18) in order to obtain the right Chern marker. The issue is then how should one consistently add a projector given that the thermal broadening at finite temperature renders the filled and empty states projectors $P$ and $Q$ in Eq. (7) rather ambiguous. For this purpose, we propose to first evaluate the matrix

$$X = \sum_{\ell<\ell'} S_\ell \hat{x} S_{\ell'} \sqrt{f_{\ell\ell'}}\,, \tag{20}$$

and the analogous $Y$ given by replacing $\hat{x} \to \hat{y}$, where $f_{\ell\ell'} \equiv f(E_\ell) - f(E_{\ell'})$. Having calculated these matrices, we define the finite temperature Chern marker by

$$\mathcal{C}^d(\mathbf{r}) = \frac{i}{a^2} \langle\mathbf{r}| \left[ X Y^\dagger - Y X^\dagger \right] |\mathbf{r}\rangle\,, \tag{21}$$

i.e., it is the diagonal element of the operator $i[X Y^\dagger - Y X^\dagger]$ that serves as the finite temperature generalization of the Chern operator defined after Eq. (8). The legitimacy of Eq. (21) relies on the fact that it encapsulates proper thermal broadening and spatially sums to the Chern number $\mathcal{C}^d = \sum_{\mathbf{r}} \mathcal{C}^d(\mathbf{r})/N$ in Eq. (18) since $\sum_{\mathbf{r}} |\mathbf{r}\rangle\langle\mathbf{r}| = I$ and $S_\ell S_{\bar\ell} = S_\ell \delta_{\ell\bar\ell}$. Essentially, our proposal is based on the assertion that the $\hat{P}\hat{x}\hat{Q}$ factor in Eq. (8) is generalized to the $X$ operator in Eq. (20) at finite temperature in order to be consistent with our linear response theory of optical conductivity, i.e., the spatial sum of the Chern marker is proportional to the global Hall conductance.

Extending Eq. 20 to the frequency-dependent matrix

$$X(\omega) = \sum_{\ell<\ell'} S_\ell \hat{x} S_{\ell'} \sqrt{f_{\ell\ell'} \delta(\omega + E_\ell/\hbar - E_{\ell'}/\hbar)} \tag{22}$$

(and the analogous $Y(\omega)$), a generalized Chern marker spectral function can now be extracted:

$$\mathcal{C}^d(\mathbf{r},\omega) = \mathrm{Re}\left[ \frac{i}{a^2} \langle\mathbf{r}| \left[ X(\omega)Y^\dagger(\omega) - Y(\omega)X^\dagger(\omega) \right] |\mathbf{r}\rangle \right]\,. \tag{23}$$

It is straightforward to see that $\mathcal{C}^d(\omega) = \sum_{\mathbf{r}} \mathcal{C}^d(\mathbf{r},\omega)/N$ ( cf. Eq. (14)). Based on Sec. 2.2 we immediately conclude that the Chern marker spectral function represents the local opacity difference at the unit cell at $\mathbf{r}$:

$$\mathcal{C}^d(\mathbf{r},\omega) = \frac{1}{8\pi\omega} \left[ \frac{\mathcal{O}^{c1}(\mathbf{r},\omega) - \mathcal{O}^{c2}(\mathbf{r},\omega)}{\pi\alpha} \right]\,, \qquad \mathcal{C}^d(\mathbf{r}) = \int_0^\infty d\omega\, \mathcal{C}^d(\mathbf{r},\omega)\,. \tag{24}$$

The local opacity sums to the global one $\mathcal{O}^{c1}(\omega) = \sum_{\mathbf{r}} \mathcal{O}^{c1}(\mathbf{r},\omega)$. As a result, $\mathcal{C}^d(\mathbf{r},\omega)$ in principle can be detected by performing the opacity measurement described after Eq. (17) locally at $\mathbf{r}$. However, at zero temperature, one should keep in mind that $\mathcal{C}^d(\mathbf{r},\omega)$ is nonzero only at frequencies larger than the band gap of the material $\omega > \Delta$. Typical semiconducting

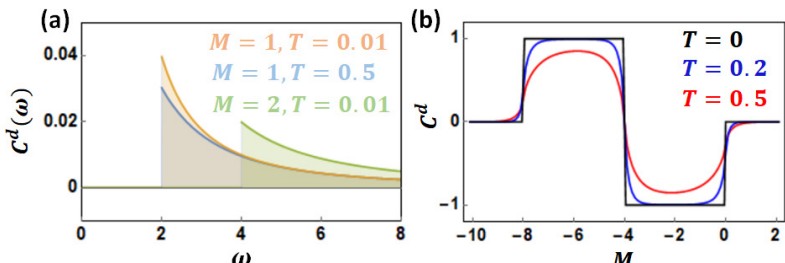

Figure 1: (a) The Chern number spectral function $\mathcal{C}^d(\omega)$ for the Chern insulator in a continuum, which is finite only at frequency larger than the bulk gap $M$, and moreover scales like $1/\omega^2$ such that it integrates to a finite value. The overall magnitude reduces with temperature. (b) The frequency-integrated Chern number $\mathcal{C}^d$ at zero and nonzero temperatures as a function of the mass term $M$.

band gaps $\Delta \sim$ eV likely necessitate circularly polarized light in the visible light range. As the wave lengths far exceed the lattice constant in this range, it will hinder the detection of local opacity in the nanometer scale. Nevertheless, we anticipate that $\mathcal{C}^d(\mathbf{r}, \omega)$ may be detected by thermal probes such as scanning thermal microscopy [41–44] that can detect the heating at the atomic scale caused by the circularly polarized light. The detected local absorption power $W_a^{c1}(\mathbf{r}, \omega) - W_a^{c2}(\mathbf{r}, \omega)$ then leads to Eq. (24) as a heating rate of the unit cell at $\mathbf{r}$.

## 3 Lattice model of Chern insulator

We now illustrate the power of the concepts described in the previous sections by exploring the concrete example of a prototypical 2D Chern insulator. The momentum space Hamiltonian in the basis $(c_{\mathbf{k}s}, c_{\mathbf{k}p})^T$ is given by [45, 46]

$$H(\mathbf{k}) = 2t \sin k_x \sigma^x + 2t \sin k_y \sigma^y + (M + 4t' - 2t' \cos k_x - 2t' \cos k_y) \sigma^z. \tag{25}$$

The internal degrees of freedom $\sigma = \{s, p\}$ are the orbitals. A straightforward Fourier transform leads to the two band lattice Hamiltonian [47]

$$
\begin{aligned}
H = \sum_i t \left\{ -i c_{is}^\dagger c_{i+ap} + i c_{i+as}^\dagger c_{ip} - c_{is}^\dagger c_{i+bp} + c_{i+bs}^\dagger c_{ip} + h.c. \right\} \\
+ \sum_{i\delta} t' \left\{ -c_{is}^\dagger c_{i+\delta s} + c_{ip}^\dagger c_{i+\delta p} + h.c. \right\} \\
+ \sum_i (M + 4t') \left\{ c_{is}^\dagger c_{is} - c_{ip}^\dagger c_{ip} \right\},
\end{aligned}
\tag{26}
$$

where $\delta = \{a, b\}$ represents the lattice constants in the two planar directions. Throughout the paper, we set $t = t' = 1.0$ and tune the mass term $M$ to examine different topological phases, and the behavior of this model at finite temperature $T$. The model hosts topological phase transitions (TPT) at three critical points $M_c = \{-8, -4, 0\}$, reflecting gap closures at different high symmetry points (HSPs) in momentum space [48]. Since they all exhibit the same critical behavior [31–33], we will focus on the $M_c = 0$ critical point where the bulk gap closes at $\mathbf{k} = (0, 0)$.

Analytical results for this model can be obtained by linearizing the Hamiltonian near the HSP $\mathbf{k}_0 = (0, 0)$, yielding $E_{m,n} = \pm\sqrt{M^2 + v^2 k^2}$ and a zero temperature Berry curvature

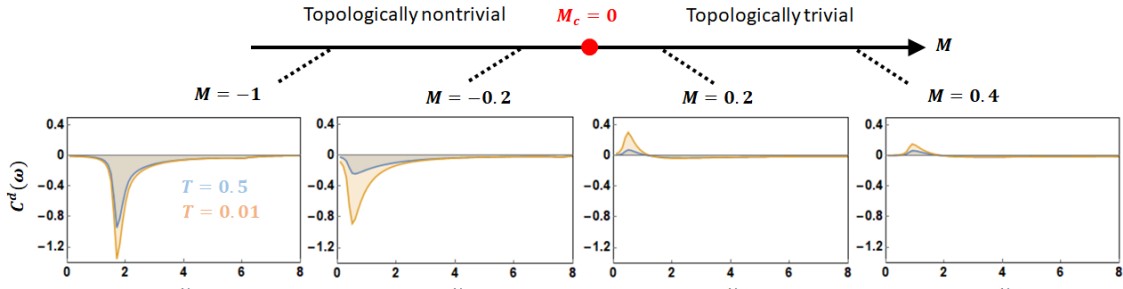

Figure 2: The Chern number spectral function $\mathcal{C}^d(\omega)$ for the lattice model of Chern insulator as a function of $M = \{-1, -0.2, 0.2, 0.4\}$ across the critical point $M_c = 0$, plotted for both low (orange) and high (blue) temperatures, where the $\delta$-function in Eq. (27) is simulated by a Lorentzian with width $\eta = 0.1$. In the topologically nontrivial phase $M = \{-1, -0.2\}$, the spectral function is negative due to the negative Chern number $\mathcal{C}^d \approx -1$, and the spectral weight gradually shifts to low frequency as $M \to M_c$. In the topologically trivial phase $M = \{0.2, 0.4\}$ and at low temperature, the positive and negative regions together yield $\mathcal{C}^d \approx 0$ (up to numerical precision).

$\Omega_{xy} = v^2 M / 2 \left( M^2 + v^2 k^2 \right)^{3/2}$. The finite temperature Chern number spectral function in Eq. (14) is given by

$$\mathcal{C}^d(\omega) = \frac{M}{2\pi\hbar\omega^2} \left[ f\left(-\frac{\hbar\omega}{2}\right) - f\left(\frac{\hbar\omega}{2}\right) \right]_{\omega \geq 2|M|/\hbar} . \tag{27}$$

At $T = 0$, $\mathcal{C}^d(\omega) \to \mathcal{C}(\omega)$ and is nonzero only if $\omega \geq 2|M|/\hbar$, since it represents an exciton absorption rate, as shown schematically in Fig. 1 (a). Moreover, the topological invariant $\mathcal{C} = \int_{2|M|/\hbar}^{\infty} d\omega\, \mathcal{C}(\omega) = \text{Sgn}(M)/4\pi$. Essentially, this is the $f$-sum rule for exciton absorption rates in circular dichroism applied to topological insulators [28]. When $T \neq 0$, since the Fermi factor $f\left(-\frac{\omega}{2}\right) - f\left(\frac{\omega}{2}\right) \leq 1$, $\mathcal{C}^d < \text{Sgn}(M)/4\pi$ is smaller than the quantized zero temperature Chern number, as illustrated in Fig. 1 (a). We anticipate that these predicted features should be readily verifiable by the opacity experiment proposed in Secs. 2.2.

The numerical results of the Chern number/marker $\mathcal{C}^d = \mathcal{C}^d(\mathbf{r})$ for the homogeneous lattice model of Chern insulator in Eq. (25) are shown in Fig. 1 (b). One sees that though the abrupt changes of the Chern number at the critical points are smeared out at nonzero temperature, clear vestiges of these TPTs are still present and should be observable in the experimentally accessible temperature range. To explain this smearing and examine the critical behavior, in Fig. 2 we present the evolution of the Chern number/marker spectral function $\mathcal{C}^d(\omega) = \mathcal{C}^d(\mathbf{r}, \omega)$ across the critical point $M_c$ at both low and high temperatures. In the topologically nontrivial phase $M < 0$, the spectral function is negative (consistent with $\mathcal{C} = -1$) and the magnitude is largest near the band gap $\omega \approx 2|M|$, reflecting that excitations of states in the vicinity of the band gap are the most detrimental to the topological properties of the system. The role of temperature is to reduce the overall magnitude of the spectral function and subsequently the frequency integration, consistent with the smearing presented in Fig. 1 (b). On the other hand, in the topologically trivial phase $M > 0$, the spectral weight has both positive and negative components such that it integrates to a zero Chern number $\mathcal{C}^d \approx 0$ at low temperature. Interestingly, the effect of temperature is to reduce the positive peak at low frequency and make the overall frequency integration slightly negative, which explains the smearing of the sharp jump of $\mathcal{C}^d$ at the critical point by temperature as shown in Fig. 1 (b). Comparing the $M = -0.2$ and $M = 0.2$ panels in Fig. 2, we see that the spectral weight near

the band gap flips sign as the system crosses the TPT at $M_c = 0$, in accordance with the flipping of Berry curvature at the HSP $\mathbf{k}_0 = (0,0)$, a defining feature of TPTs [31–33,49]. Interestingly, these features of $\mathcal{C}^d(\omega)$ bear a striking similarity with the Haldane-type Floquet topological insulator [22]. The latter has been realized in cold atoms that has an extremely narrow band width $\sim 10^{-12}$eV, in which the zero temperature limit $\lim_{T \to 0} \mathcal{C}^d(\omega)$ hs been measured. This indicates that these features may be generic for Chern insulators realized in a variety of different energy scales.

Finally, combining the shape of $\mathcal{C}^d(\omega)$ in Fig. 2 with the opacity measurement proposed in Sec. 2.2 implies a remarkably simple way to infer the finite temperature Chern number in 2D materials. Figure 2 suggests that if a 2D material always appears more transparent under right circularly polarized light than the left (or vice versa) at any frequency, then the material must be topologically nontrivial, as $\mathcal{C}^d(\omega)$ is always of the same sign and hence it must frequency-integrate to a finite Chern number. Depending on the frequency range of $\mathcal{C}^d(\omega)$ in real materials, this should be directly visible to the naked eye or through an infrared/UV lens, offering a very simple way to perceive the topological order in the macroscopic scale. On the other hand, if the transparency of the material under the two circular polarizations is strongly frequency dependent, then a frequency integration of $\mathcal{C}^d(\omega)$ is required to infer the Chern number.

# 4 Topological quantum criticality

The Chern marker is known to display interesting critical behavior near TPTs, such as size-dependent smoothening of its discontinuity [50], Kibble-Zurek scaling in disordered Chern insulators [51], and Hofstadter-butterfly-like features in quasicrystals [52]. As with standard symmetry breaking critical points, where correlation functions of the order parameter show divergent correlation lengths, here, using linear response theory, we explore if there exist certain nonlocal correlators that will display such singular behavior near TPTs. We identify two quantities: a *Chern correlator* and a *nonlocal Chern marker*, which encode different physics pertaining to the topological quantum criticality.

## 4.1 Chern correlator

Based on the linear response theory presented earlier, we define a Chern correlator spectral function by splitting the second position operator $\hat{x} = \sum_{\mathbf{r}'} \hat{x}_{\mathbf{r}'}$ in Eq. (23) into its component on each site $\mathbf{r}'$, yielding

$$
\begin{aligned}
\tilde{\mathcal{C}}(\mathbf{r}, \mathbf{r}', \omega) &= \mathrm{Re}\left[ \frac{i}{a^2} \langle \mathbf{r} | \left[ X(\omega) Y_{\mathbf{r}'}^\dagger(\omega) - Y(\omega) X_{\mathbf{r}'}^\dagger(\omega) \right] | \mathbf{r} \rangle \right], \\
X_{\mathbf{r}'}^\dagger(\omega) &= \sum_{\ell < \ell'} S_{\ell'} \hat{x}_{\mathbf{r}'} S_\ell \sqrt{f_{\ell\ell'}} \delta(\omega + E_\ell/\hbar - E_{\ell'}/\hbar),
\end{aligned}
\tag{28}
$$

which spatially sums to the Chern marker spectra function $\mathcal{C}^d(\mathbf{r}, \omega) = \sum_{\mathbf{r}'} \tilde{\mathcal{C}}(\mathbf{r}, \mathbf{r}', \omega)$ in Eq. (23). This splitting of the second position operator is justified by the observation that in Eqs. (11) and (18), the second position operator accounts for the global field. Consequently, this Chern correlator spectral function $\tilde{\mathcal{C}}(\mathbf{r}, \mathbf{r}', \omega)$ represents the local current at site $\mathbf{r}$ caused by applying a field of frequency $\omega$ at site $\mathbf{r}'$, i.e., the nonlocal current response. A frequency

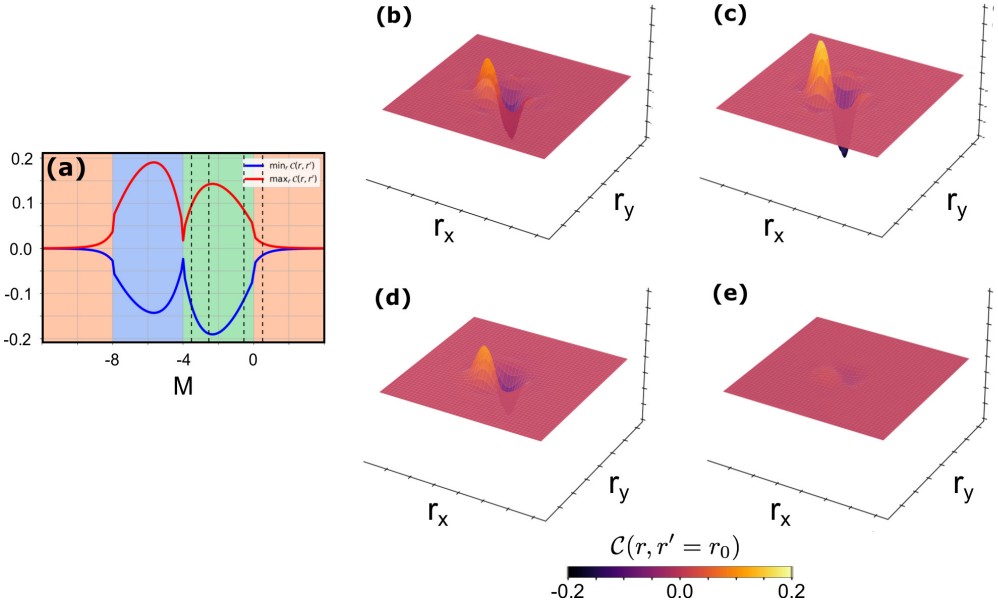

Figure 3: (a) The maximal and minimal values of the zero-temperature Chern corre-lator $\tilde{\mathcal{C}}(\mathbf{r}, \mathbf{r}')$ as a function of the tuning parameter $M$ for a Chern insulator on a $14 \times 14$ lattice. The background color indicates the three topologically distinct phases. The vertical dashed lines indicate the surface plots of $\tilde{\mathcal{C}}(\mathbf{r}, \mathbf{r}' = \mathbf{r_0})$ with $r_0 = (7, 7)$, shown in the right panels: (b) $M = -3.5$, (c) $M = -2.5$, (d) $M = -0.5$, (e) $M = 0.5$. The surface plots have been interpolated on a $140 \times 140$ grid as a visual aid.

integration $\tilde{\mathcal{C}}(\mathbf{r}, \mathbf{r}') = \int_0^\infty d\omega\, \tilde{\mathcal{C}}(\mathbf{r}, \mathbf{r}', \omega)$ further leads to a Chern correlator

$$
\begin{aligned}
\tilde{\mathcal{C}}(\mathbf{r}, \mathbf{r}') &= \mathrm{Re}\left[ \frac{i}{a^2} \langle \mathbf{r} | \left[ X\, Y_{\mathbf{r}'}^\dagger - Y\, X_{\mathbf{r}'}^\dagger \right] | \mathbf{r} \rangle \right], \\
X_{\mathbf{r}'}^\dagger(\omega) &= \sum_{\ell < \ell'} S_{\ell'} \hat{x}_{\mathbf{r}'} S_\ell \sqrt{f_{\ell\ell'}}.
\end{aligned}
\tag{29}
$$

This correlator sums to the Chern marker $\sum_{\mathbf{r}'} \tilde{\mathcal{C}}(\mathbf{r}, \mathbf{r}') = \mathcal{C}^d(\mathbf{r})$ and represents a measure of the internal fluctuation of the Chern marker. Note that in a clean and infinite system, the local Chern marker $\mathcal{C}^d(\mathbf{r}) = \mathcal{C}^d$ is homogenous even in the vicinity of a TPT. However, the Chern correlator $\tilde{\mathcal{C}}(\mathbf{r}, \mathbf{r}')$ depends on $\mathbf{r} - \mathbf{r}'$ as clearly evidenced by the numerical results for the Chern insulator shown in Fig. 3.

Typically, the correlations are maximally deep in a topological phase and very weak in topologically trivial phases, indicating a much larger internal fluctuation of the Chern marker in the nontrivial phase. The Chern correlator consists mainly of two peaks of opposite sign around $\mathbf{r} \sim \mathbf{r}'$ consistent with the expectation that the nonlocal current-current correlator is larger at short separations. A pronounced asymmetry between the heights of the two peaks is seen in topologically non trivial phases whereas the peak heights are equal in the topologically trivial phase, concomitant with the expectation that the spatial integral of this correlator should yield the Chern number. These aspects are well encapsulated in Fig. 3(a) where we see the evolution of the two peak heights as a function of $M$. Finally, the frequency dependence of the Chern correlator spectral function $\tilde{\mathcal{C}}(\mathbf{r}, \mathbf{r}', \omega)$ is shown in Fig. 4, indicating a spatial pattern that changes dramatically with $\omega$. Physically, this reflects the variation of the current at $\mathbf{r}$ stemming from optical absorption due to the light applied at $\mathbf{r}'$.

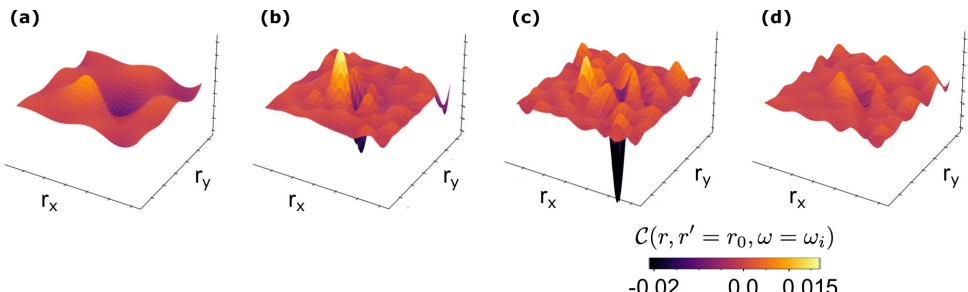

Figure 4: Chern correlator spectral function $\tilde{C}(\mathbf{r}, \mathbf{r}' = r_0, \omega = \omega_i)$ at various values of the frequency $\omega_i$ for a Chern insulator at $M = -1.0$ and temperature $k_B T = 0.1$ on a $12 \times 12$ lattice. The reference point is $r_0 = (6,6)$. (a) $\omega_i = 2.7$, (b) $\omega_i = 5.6$, (c) $\omega_i = 5.9$, (d) $\omega_i = 6.9$. In the numerics, the delta function was approximated by a Lorentzian with width parameter $\eta = 0.1$. The surface plots have been interpolated on a $120 \times 120$ grid as a visual aid.

## 4.2  Wannier state correlation function as a nonlocal Chern marker

Lastly, we highlight the intricate link between Chern markers and a previously proposed Wannier state correlation function [31–33]. Restricting ourselves to a homogeneous system at zero temperature, the Wannier state correlation function derived from the Fourier transform of the Berry curvature $\Omega_{xy}$ is [36, 37, 53]

$$
\tilde{F}(\mathbf{R}) = \int \frac{d^2\mathbf{k}}{(2\pi)^2} \Omega_{xy}(\mathbf{k}) e^{i\mathbf{k}\cdot\mathbf{R}} = -i \sum_n \langle \mathbf{R}n | (R_x \hat{y} - R_y \hat{x}) | \mathbf{0}n \rangle
$$

$$
= -i \sum_n \int d^2\mathbf{r} (R_x y - R_y x) W_n(\mathbf{r} - \mathbf{R})^* W_n(\mathbf{r}), \tag{30}
$$

provided $\mathbf{R} \neq \mathbf{0}$. The Lorentzian shape of the Berry curvature $\Omega_{xy}(\mathbf{k})$ with width $\xi^{-1}$ results in a decaying $\tilde{F}(\mathbf{R})$ with correlation length $\xi$. The spatial profile of $\tilde{F}(\mathbf{R})$ for the lattice Chern insulator of Eq. (26) is discussed in Fig. 3 of Ref. [31]. As the system approaches the TPTs, $\xi \to \infty$. This is a generic feature of TPTs both in and out of equilibirum [54, 55], from which the critical exponent of $\xi$ can be extracted [31–33].

Using the formalism in Sec. 2.1, we also note that

$$
\tilde{F}(\mathbf{R}) = \sum_n \int \frac{d^2\mathbf{k}}{(2\pi)^2} i \langle \partial_x n^{\mathbf{k}} | \partial_y n^{\mathbf{k}} \rangle e^{i\mathbf{k}\cdot\mathbf{R}} - (x \leftrightarrow y)
$$

$$
= \sum_n \sum_m \int \frac{d^2\mathbf{k}}{(2\pi)^2} \int \frac{d^2\mathbf{k}'}{(2\pi)^2} i \langle \psi_{n\mathbf{k}} | \hat{x} | \psi_{m\mathbf{k}'} \rangle \langle \psi_{m\mathbf{k}'} | \hat{y} | \psi_{n\mathbf{k}} \rangle e^{i\mathbf{k}\cdot\mathbf{R}} - (x \leftrightarrow y). \tag{31}
$$

Since $\mathbf{R}$ is a Bravais lattice vector and $n^{\mathbf{k}}(\mathbf{r}) = n^{\mathbf{k}}(\mathbf{r} + \mathbf{R})$ is cell periodic,

$$
\langle \mathbf{r} | \psi_{n\mathbf{k}} \rangle e^{i\mathbf{k}\cdot\mathbf{R}} = e^{i\mathbf{k}\cdot(\mathbf{r}+\mathbf{R})} n^{\mathbf{k}}(\mathbf{r}+\mathbf{R}) = \psi_{n\mathbf{k}}(\mathbf{r}+\mathbf{R}) = \langle \mathbf{r}+\mathbf{R} | \psi_{n\mathbf{k}} \rangle. \tag{32}
$$

In parallel, using the projector operator formalism, we can define a nonlocal Chern marker $C(\mathbf{r}+\mathbf{R}, \mathbf{r})$ from the off-diagonal element of the Chern operator

$$
C(\mathbf{r}+\mathbf{R}, \mathbf{r}) \equiv \mathrm{Re}\left[ \frac{i}{a^2} \langle \mathbf{r}+\mathbf{R} | \left[ \hat{P}\hat{x}\hat{Q}\hat{y}\hat{P} - \hat{P}\hat{y}\hat{Q}\hat{x}\hat{P} \right] | \mathbf{r} \rangle \right]. \tag{33}
$$

Note that the case $\mathbf{R} = 0$ reduces to the standard Chern marker $C(\mathbf{r})$. Using Eq. (32), one sees that for homogeneous systems in the thermodynamic limit, the nonlocal Chern marker is precisely the $\tilde{F}(\mathbf{R})$

$$
\lim_{N\to\infty} C(\mathbf{r}+\mathbf{R}, \mathbf{r}) = \tilde{F}(\mathbf{R}). \tag{34}
$$

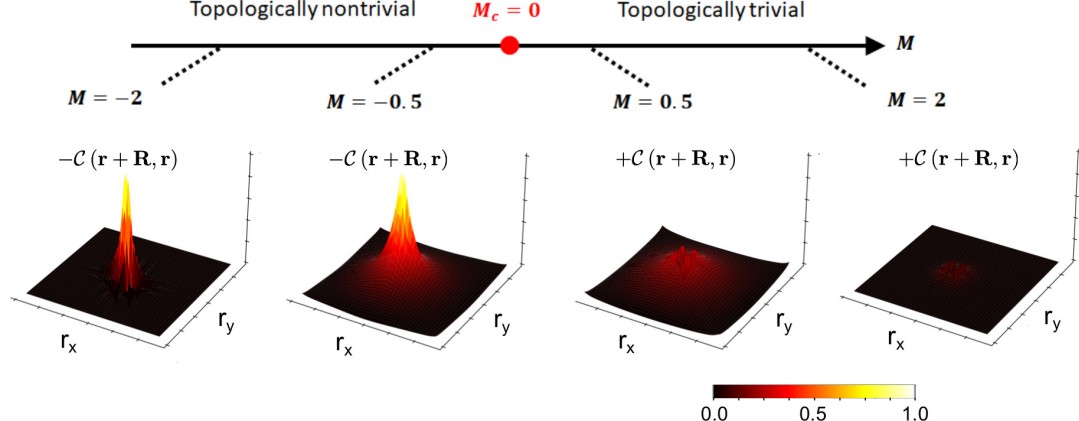

Figure 5: The nonlocal Chern marker $\mathcal{C}(\mathbf{r}+\mathbf{R},\mathbf{r})$ of the lattice Chern insulator of size $28 \times 28$ sites, where we choose $\mathbf{r}$ to be at the center of the lattice and plot it as a function of $\mathbf{R}$ for four values of $M$. One sees that the spatial profiles closer to the critical point $M = \pm 0.5$ are more long-ranged than those far away from the critical point $M = \pm 2$. Note that we plot $-\mathcal{C}(\mathbf{r}+\mathbf{R},\mathbf{r})$ for the topologically nontrivial phase since the Chern marker $\mathcal{C}(r) \approx -1$ gives an overall negative amplitude.

This remarkable correspondence offers a highly practical method to evaluate the Wannier state correlation function $\tilde{F}(\mathbf{R})$ in lattice models. The spatial profile of $\mathcal{C}(\mathbf{r}+\mathbf{R},\mathbf{r})$ for the lattice Chern insulator in Eq. (26) for parameters close and far away from the critical point $M_c = 0$ is shown in Fig. 5. As expected, $\mathcal{C}(\mathbf{r}+\mathbf{0},\mathbf{r}) = \mathcal{C}(\mathbf{r}) \approx 0$ or $-1$ in the two distinct topological phases and $\mathcal{C}(\mathbf{r}+\mathbf{R},\mathbf{r})$ decreases with $\mathbf{R}$. Due to the diverging correlation length $\xi$, $\mathcal{C}(\mathbf{r}+\mathbf{R},\mathbf{r})$ becomes more long-ranged as the system approaches the critical point $M_c = 0$ from either side of the transition, thereby serving as a faithful measure to the critical behavior near TPTs. We remark that the critical exponent of the correlation length $\xi \sim |M|^{-\nu}$ has been calculated previously from the divergence of Berry curvature, yielding $\nu = 1$ for linear Dirac models [31–33,54–56]. Similar critical exponents can also be extracted from the scaling of the Chern marker itself [50] These features remain true for all the three critical points $M_c = \{0, -4, -8\}$ in the lattice model of Chern insulators, so we only present the data near the critical point $M_c = 0$ for simplicity. Though $\mathcal{C}(\mathbf{r}+\mathbf{R},\mathbf{r})$ does not have a straightforward interpretation within linear response theory, it can nevertheless be extracted from Fourier transforms of the Berry curvature measured in momentum space. Importantly, as the decay length $\xi$ is entirely determined by the Lorentzian shape of the Berry curvature at HSPs [31,33], it suffices to probe a very small region near the HSPs to extract the decay of the Wannier correlation. Such measurements should be feasible with pump-probe experiments in 2D materials [27,57].

## 5 Conclusions

In summary, we unveil a number of new aspects related to the measurement of the Chern number and the identification of TPTs in 2D TR breaking materials as summarized schematically in Fig. 6. We first identify a Chern number spectral function $\mathcal{C}^d(\omega)$ that can be extracted from the opacity of 2D materials against circularly polarized light. As the Chern number $\mathcal{C}^d$ is simply given by the frequency integration of this spectral function, our results provide a very simple experimental protocol to access bulk topology that is expected to be widely applicable to 2D materials. Generalizing the above measurement protocol to finite temperatures, we show that

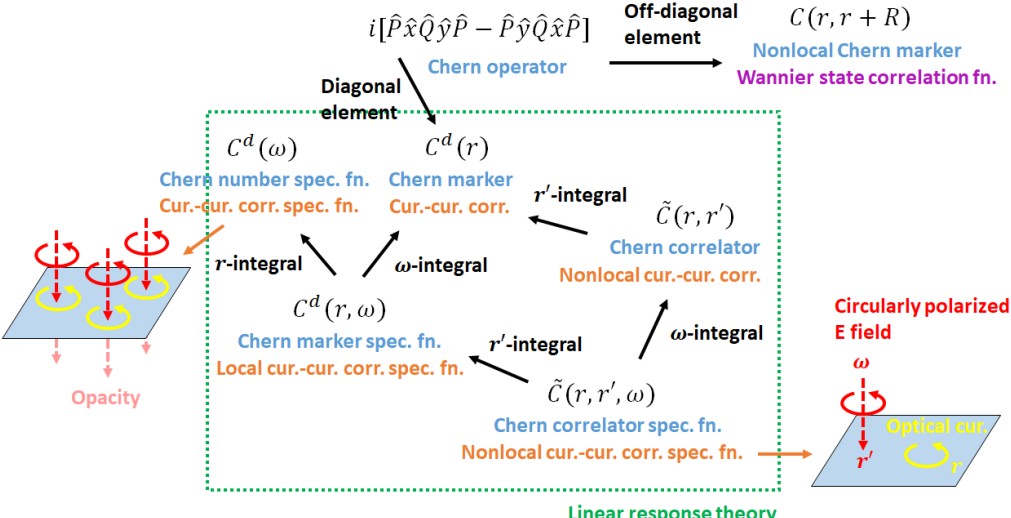

Figure 6: Summary of the linear response theory, measurement protocols, and Wannier function interpretation of various quantities introduced in the present work. The blue text gives the terminology, the orange text the corresponding physical quantity in the optical Hall response, and the black arrows the relation between them. The abbreviations are cur.-cur. corr. = current-current correlator, spec. = spectral, and fn. = function.

temperature adversely affects the low frequency part of $\mathcal{C}^d(\omega)$ thereby smearing the sharp jumps of the Chern number at zero temperature TPTs. We then propose a finite temperature Chern marker spectral function $\mathcal{C}^d(\mathbf{r}, \omega)$ which frequency-integrates to the finite temperature Chern marker. This spectral function can be measured by atomic scale thermal probes as the local heating rate caused by circularly polarized light.

To quantify critical behavior near TPTs in 2D TR breaking materials, we introduced the Chern correlator $\tilde{\mathcal{C}}(\mathbf{r}, \mathbf{r}')$ and its associated spectral function $\tilde{\mathcal{C}}(\mathbf{r}, \mathbf{r}', \omega)$. These are intimately linked to nonlocal current-current correlations in the optical Hall response. These quantities manifest different behaviors in topologically trivial and non-trivial phases. We also showed that the nonlocal Chern marker becomes increasingly long-ranged as TPTs are approached. Finally, we remark that the notions introduced in the present work have been recently generalized to TR symmetric 2D systems like the Bernevig-Hughes-Zhang model [58] described by spin Chern numbers and markers [59]. We anticipate that these new aspects may be widely applied to investigate the influence of real space inhomogeneities on both bulk topology and the critical behavior near TPTs.

# 6  Acknowledgments

We thank P. d'Ornellas and R. Barnett for stimulating discussions.

**Funding information** This work is partially supported by the ESPRC Grant no. EP/P009565/1. W. C. acknowledges the financial support of the productivity in research fellowship from CNPq. We also acknowledge computation time on the ETH Euler cluster.

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
