# Peer review of "Probing Chern number by opacity and topological phase transition by a nonlocal Chern marker"

_SciPost Physics Core, doi:SciPost Phys. Core 6, 059 (2023)_

## Round 2 · Referee Report · Anonymous (Referee 1) · 2022-9-10

Strengths

1- The paper provides a useful conceptual extensions of the Chern marker formalism, demonstrating that it can be recast in terms of response functions of the system to circularly polarised light. Specifically, the authors define a Chern marker correlator, a Chern marker spectral function, and a non-local generalisation of the Chern marker 2- The interpretation of these quantities as response functions provides a straightforward generalisation of the quantities to finite temperature. 3- The proposed quantities are experimentally measurable by probing the response to circularly polarised light.

Weaknesses

1- The data presented in the paper show a few numerically evaluated snapshops of the new quantities, but fall short of a systematic study. As a result, the paper cannot make strong conclusions about the general usefulness of these concepts.

Report

This is a well written paper, which introduces new concepts in the study of real-space Chern markers.

The work provides new tools to investigate topological bands, which I believe satisfies an essential criterion for this journal, to "Open a new pathway in an existing or a new research direction, with clear potential for multipronged follow-up work".

The potential for an experimental measurement of the proposed quantities is also promising for follow-up work.

I recommend the paper for publication, subject to minor changes. The authors may with to improve the paper by including a more systematic discussion of the different Chern response functions in different regimes.

Requested changes

1- The Chern insulator model given in equation (29) looks like a version of the two-orbital square lattice model known in the literature on (fractional) Chern insulators, corresponding to the spin polarised version of the model introduced in B. A. Bernevig, T. L. Hughes, and S.-C. Zhang, Science 314, 1757 (2006). An original reference should be included. 2- The authors note on page 13 that "the critical exponent of the correlation length ξ ∼ |M|−ν has been calculated previously from the divergence of Berry curvature". Given that the current paper is about Chern markers, it seems appropriate to also mention that the correlation length exponent has indeed been measured from the behaviour of the Chern marker also, e.g. in Ref. 38. 3- Consider adding more systematic exploration of the different Chern response functions.

---

## Round 2 · Referee Report · Anonymous (Referee 2) · 2022-10-28

Strengths

  1. This work introduces generalized physical quantities derived from the so-called "local Chern marker" (introduced by Bianco and Resta [7]), which could be used to signal phase transitions in Chern insulators.

  2. These generalizations have the potential to introduce novel probes and concepts in the field of topological quantum matter.

Weaknesses

  1. Most of the introduced markers are equivalent to well-known quantities (e.g. the nonlocal Chern marker in Eq. 38 is equivalent to the Wannier-state correlation function of Ref. 25; the "generalized nonzero temperature local Chern marker" in Eq. 23 seems to simply correspond to the local Hall response at finite temperature). In this sense, it is not clear whether these "generalized markers" are in fact useful/novel.

  2. Related to the point above: the experimental access to these "generalized markers" is not properly described in the manuscript. What are the probing fields and detection tools that are required to access these markers in experiments?

  3. In general, the informations provided by these generalized markers (such as the "topological correlations" associated with the "Chern correlator") remain vague.

  4. The manuscript contains a series of vague and/or misleading statements.

Report

This work builds on the concept of "local Chern marker" (introduced by Bianco and Resta [7]) to propose a series of generalized physical quantities, which could be used to signal topological phase transitions in Chern insulators. While this work makes interesting links between different quantities (markers, response functions, correlation functions, ... ), it is not clear whether the proposed markers provide any new, concrete, or relevant information on topological states and their phase transitions. In this sense, I find the scope of this work rather limited.

Here are a series of concrete remarks:

  • In the introduction, I found the following sentence misleading: "The bulk band structure, on the other hand, shows the same gapped energy spectrum in both the topologically trivial and nontrivial phases, and therefore bulk measurements not involving edge states are often not considered feasible to identify the Chern number". This statement seems to contradict the well-known TKNN result, according to which the Hall conductivity (as obtained from Kubo's formula, considering a system with periodic boundary conditions) is related to the Chern number. From the TKNN result, it is clear that the Chern number can be identified from "bulk measurements not involving edge states", which is in sharp contrast with the authors' statement. In this sense, I believe that this statement should be revised.

  • Typo on page 2: "which is greatly improves their relevance"

  • Page 3: the sentence "To relate the Chern number to our proposed Chern marker" introduces an ambiguity: here the authors seem to refer to the well-known Bianco-Resta marker, and not to their new markers (introduced later in Section 4). Similarly, a clear citation to Bianco and Resta [7] is needed above Eqs. 9 and 12. [Note: Eq. 12 should end with a full stop '.'].

  • I was puzzled by the opening of Section 2.3 "The result linking the Chern number to the circular dichroism can be generalized to nonzero temperatures using linear response theory", which seems to announce the derivation of a fundamentally new result. As far as I can see, it is trivial to generalize Eqs. 15-16 to finite temperature, by simply inserting a Boltzmann weight [exp (- E_n/kT)] inside the sum of Eq. 15. By doing so, the right-hand side of Eq. (16) becomes the finite-temperature Hall conductivity, which (as far as I can see) is equivalent to the quantity displayed in Eq. (23) and Fig. 1(d). Similarly, the quantity shown in Figs. 1 (a)-(c) seems to correspond to the differential rate [the integrand in Eq. 16] at finite temperature, which is simply obtained by inserting the Boltzmann weight in Eq. 15. In this sense, I could not identify the novelty (or non-triviality) here.

  • About the quantity displayed in Eq. 23, the authors wrote "To summarize, Cd(r) is a real space generalization of a topological probe which is intimately related to the measurable charge polarization susceptibility". I found this statement misleading: indeed, it seems to me that the quantity in Eq. 23 simply corresponds to the local Hall response of the system at finite temperature. Altogether, I could not identify the novelty of the results/quantities presented in that Section 2.3.

  • Similarly, I found the following sentence misleading (also below Eq. 23): "The generalization to nonzero temperature is an important milestone to address topology in more realistic scenarios". The notion of finite-temperature Hall response is well known, and in this sense, I believe that this is not a "new generalization" nor "an important milestone".

  • I would invite the authors to explain why the alternative expressions in Eq. 27-28 are in fact useful.

  • Below Eq. (29), I found the statement "represents the shift to the neighbouring sites" unclear. Besides, a sketch of this model would have been useful.

  • Below Eq. (30): I found the following statement misleading "and tune the mass term M and temperature kBT to examine different topological phases", which seems to suggest (implicitly) that varying temperature could lead to different topological phases.

  • Above Eq. (31): The sentence "The Chern number and Chern marker spectral functions are equivalent in this homogeneous model, giving ... ", together with the following Eq. 31, seems to suggest that the Chern number is a well-defined topological invariant at finite temperature, which is misleading.

  • Below Eq. (32): The authors wrote "This Chern correlator represents a measure of topological correlation in the system" and then later "the Chern correlator (...) can therefore act as a proxy for the amount of spatial topological correlation in the system". Could the authors rigorously define the concept of "topological correlation"? What is topological about these correlations? Why should we care about these correlations (what new/specific informations do they provide)?

  • About Fig. 4, the authors wrote "Note that the spectral function exhibits a rich spatial structure, indicating that the correlation within the system can vary a lot depending on the probed frequency". I had difficulties identifying any concrete information that one could get from that result (and the related statement in the text). What do we learn from this rich spatial structure?

  • In the concluding paragraph, the authors wrote "All of these quantities can be measured in appropriately tailored circular dichroism experiments." I found that statement very vague.

  • In general: It would be useful to clarify (i) what are the physical quantities that are genuinely new in this work, (ii) how they could be measured in practice (what probing field, what detection technique is needed?), and (iii) what new informations/advantages do these new quantities offer (as compared to well-known quantities such as Hall responses, dichroic signals, Fourier-transformed Berry curvatures, ... ).

Requested changes

See report above.

---

## Round 4 · List of Changes

(1) In the beginning of Sec 2.2, we have mentioned that our formalism is analogous to that derives the Faraday effect in solids.
(2) At the end of Sec 2.2, we have elaborated explicitly that our finite temperature formalism that describes the optical absorption process is different from the finite temperature formalism that describes the DC Hall conductance.
(3) In page 10, we have mentioned the experimental measurement of the Chern number spectral function in cold atoms.
(4) In page 10 and 11, we demonstrate a simple way to perceive the topologically nontrivial phase of a 2D time-reversal breaking material by human eyes from the transparency of the material in macroscopic scale.

---

## Editorial Decision

published